

# Assessment of the contribution of residential waste burning to ambient PM$_{10}$ concentrations in Hungary and Romania

András Hoffer[1,2], Aida Meiramova[2], Ádám Tóth[2], Beatrix Jancsek-Turóczi[1,2], Gyula Kiss[3], Erika Andrea Levei[4], Luminita Marmureanu[5,6], Attila Machon[7], András Gelencsér[1,2]

5  [1]ELKH-PE Air Chemistry Research Group, Veszprém, H-8200 Hungary

[2]Research Institute of Biomolecular and Chemical Engineering, University of Pannonia, Veszprém, H-8200, Hungary

[3]Renewable Energy Research Group, University of Pannonia Nagykanizsa - University Center for Circular Economy, Nagykanizsa, 8800, Hungary

10  [4]Research Institute for Analytical Instrumentation Subsidiary, INCDO-INOE 2000, Cluj-Napoca, RO-400293, Romania

[5]National Institute of Research and Development for Optoelectronics INOE2000, 409 Atomiştilor, RO-077125, Măgurele, Ilfov, Romania

[6]National Institute for Research and Development in Forestry "Marin Drăcea" - INCDS, Voluntari, RO-077030, 15 Romania

[7]Air Quality Reference Centre, Hungarian Meteorological Service, Budapest, H-1181, Hungary

*Correspondence to:* András Hoffer (hoffer.andras@mk.uni-pannon.hu)

**Abstract.** The illegal burning of solid wastes in residential stoves is an existing practice yet until now it has been completely disregarded as an emission source of atmospheric pollutants in many developed countries including 20 those in Eastern Europe. Various types of solid wastes (plastics, treated wood, plyboards, tyre, rag, etc.) serve as an auxiliary fuel in many households, in particular during the heating season. In this work, for the first time ever in atmospheric pollution studies, specific tracer compounds identified previously in controlled test burnings of different waste types in the laboratory were detected and quantified in ambient PM$_{10}$ samples collected in 5 Hungarian and 4 Romanian settlements. Using the identified tracers and their experimentally determined relative 25 emission factors the potential contribution of illegal waste burning emissions to ambient PM$_{10}$ mass concentrations was assessed. Our findings implied that the burning of PET-containing waste (food and beverage packaging, clothes) was predominant at all locations, especially in North-Eastern Hungary and Romania. There is substantial evidence that the burning of scrap furniture is also common in big cities in Hungary and Romania. Back-of-the-envelope calculations based on the relative emission factors of individual tracers suggested that the 30 contribution of solid waste burning particulate emissions to ambient PM$_{10}$ mass concentrations may be as high as a few percents. This finding, when considering the extreme health hazards associated with particulate emissions from waste burning, is a matter of serious public health concerns.

## 1 Introduction

Burning of solid fuels in households is a significant source of atmospheric particulate matter and gaseous 35 pollutants worldwide. In most countries in Europe the predominant type of solid fuel in households is fuel wood. Using the tracer approach based on the cellulose pyrolysis product levoglucosan Caseiro et al. (2009) estimated that the contributions of wood burning to the PM$_{10}$ concentrations were 10 and 20% in Vienna and in rural settlements in Austria, respectively. In Budapest, up to 40% of the carbon in PM$_{10}$ was found to be emitted from wood burning in winter (Salma et al., 2017). The organic carbon emitted from biomass burning represented



about 80% of the $PM_1$ carbon in winter near Bucharest as determined by a compact time-of-flight aerosol mass spectrometer (Marmureanu et al., 2020).

Besides biomass burning, the burning of different types of household wastes is also an important emission source of particulate matter worldwide (Christian et al., 2010; Wiedinmyer et al., 2014; Kumar et al., 2015). Open waste burning is quite common in many countries, especially in which organized waste collection systems

are lacking, costly, or collection service is infrequent. Emissions from modern waste incinerator plants are incomparably lower than from open waste burning or from the burning of solid wastes in household stoves (Lemieux et al., 2004; Jones and Harrison 2016). Based on the high concentrations of phthalic acid and bisphenol-A in urban particulates Kanellopoulos et al. (2021) suggested that the burning of plastic wastes is a non-negligible source of air pollution especially in the autumn/winter season in the industrial district of

Aspropirgos, Greece. Wiedinmyer et al. (2014) estimated the amount of waste burned in households and at dump sites based on the general guidelines reported in IPCC 2006. For Hungary and Romania, they found that the amount of the waste burned on an annual basis was about 2% and 60% of the generated waste of the given country, respectively. The estimated contributions of waste burning to the ambient $PM_{10}$ were 4 and 35% in Hungary and Romania, respectively.

Being an illegal and uncontrolled practice the assessment of the magnitude of residential waste burning and its effect on air quality poses an extreme challenge to environmental authorities. Apart from the fact that the amount of solid wastes produced in households can only be estimated with a high level of uncertainty, the fraction which is burned in residential stoves is practically unknown. Quite recently two independent surveys were conducted targeting the waste burning practices in Hungary. The survey organized by Kantar Hoffman Ltd. found that

about 10% of the pollees admitted to burn solid waste on a regular basis, mostly plastics, treated wood and clothes. A telephone-based survey conducted by the Századvég Foundation in 2018 concluded that 4% of the population burned household wastes (treated wood, rag, paper, plastics) indoor, nearly half of them on a daily basis.

Hoffer et al. (2021) measured the emission factors of $PM_{10}$ and particle-bound PAHs emitted upon burning of

various types of solid wastes in a residential stove in the laboratory. Potential tracer compounds specific to different waste types were also identified and their emission factors were determined. 1,3,5-triphenylbenzene is considered to be a non-specific tracer for the low-temperature burning of several types of solid waste (polystyrene (PS), polyethylene terephthalate (PET), coated paper, furniture panles, acrylonitrile butadiene styrene polymer (ABS)) and has been identified in atmospheric particulates (Simoneit et al., 2005; Kumar et al.,

2015; Zhao et al., 2018; Furman et al., 2020). However, there has been no systematic study on the estimated mass contribution of solid waste burning particulate emissions to ambient $PM_{10}$ concentrations. To the best of our knowledge, our work is the very first study that attempts to quantitatively assess the magnitude of residential waste burning particulate emissions and their contribution to atmospheric $PM_{10}$ concentrations in different settlements based on the laboratory measurements of the relative emission factors of highly specific tracers of

waste burning. Our approach is analogous to the source apportionment of biomass burning in atmospheric aerosol using levoglucosan as a tracer.

## 2 Material and methods

### 2.1 $PM_{10}$ filter sampling



Sampling was performed at 5 Hungarian and 4 Romanian settlements, their geographic locations are shown in
Figure 1.

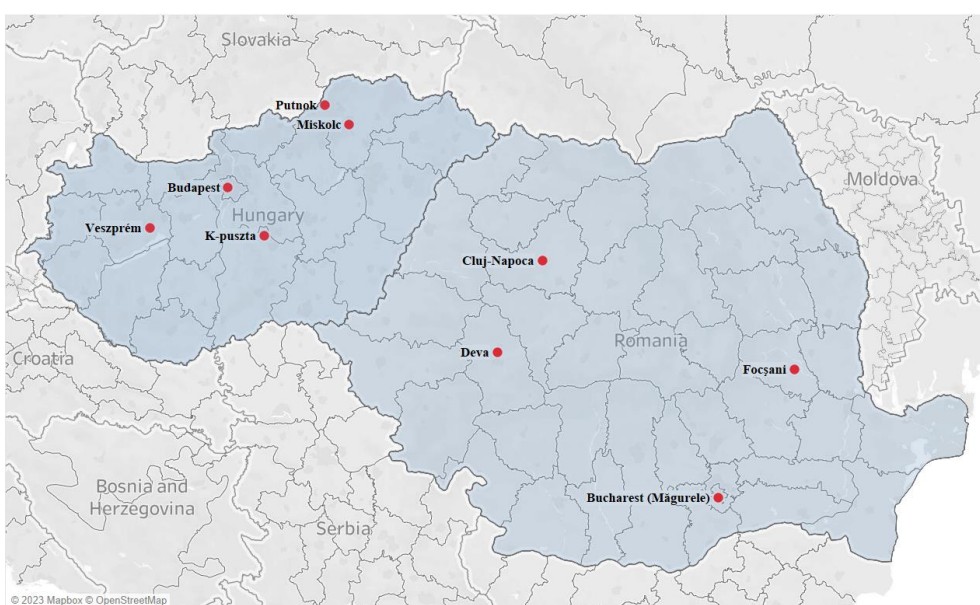

**Figure 1 Geographic locations of the sampling sites.**

The measurement station in Miskolc (MSK) was located in the city centre (Búza Square), near two busy roads,
and classified as a traffic-impacted site. Putnok (PUT) is a small town located in the Sajó valley which is
reportedly the most polluted region in Hungary with respect to the number of exceedances of the health limit for
$PM_{10}$ mass concentrations. Here the sampling station was located inside the courtyard of the High School for
Agriculture, the sampling site is classified as rural. In the city of Budapest (BUD) the sampling was performed in
the garden of the György Marczell Observatory of the Hungarian Meteorological Service, located 14 km to the
south-east from the city centre, in the middle of a suburban residential area. K-puszta (KPS) is a regional
background air quality monitoring station located in a forest clearing 15 kilometres from the city centre of
Kecskemét. In Veszprém (VES) $PM_{10}$ samples were collected in a residential area in a small river valley close to
the city centre (Patak Square). In Deva (DEV) the sampling site was downtown in the yard of a company, in the
Dragos Voda Street, at about 10 km from a coal and natural gas fired power station. In Cluj-Napoca (CLJ)
samples were collected in the yard of the Research Institute for Analytical Instrumentation in the western part of
the city (Donath Street), near two busy roads in the immediate vicinity of the suburban air quality monitoring
station of the Romanian Environmental Protection Agency. In Focsani (FOC) the sampling was performed in the
backyard of a private house, in a suburban residential area. In Bucharest $PM_{10}$ samples were collected in the
metropolitan area Măgurele, which is located about 12 km from the city centre to the south-west. In Măgurele
the $PM_{10}$ samples were collected at two sampling stations, one was located in the yard of the National Institute
for Research and Development for Optoelectronics - INOE 2000 (BUC-R), and the other station (BUC-M) was
located 2 km away from the first station (Atmosferei Street), and was used only during the winter of 2020.




At all stations the PM$_{10}$ samples were collected on quartz filters (Advantec QR-100) with Digitel DHA-80 high
volume samplers. The sampling time was 24 hours starting from midnight for each sample. Before and after the
sampling the filters were conditioned in a weighting room at 20 ± 1 °C and at a relative humidity of 45−50% for
3 days. The PM$_{10}$ mass concentrations were determined according to the EN 12341:2014 Ambient air-Standard
gravimetric measurement method.

PM$_{10}$ filter samples were collected during the heating season of 2018/2019 and 2019/2020 and also in the
110 summer of 2019. Table 1 summarises the sampling dates, the number of the samples analysed and the average
PM$_{10}$ concentrations obtained from the gravimetric analyses.

| Location | Sampling period | Number of samples analysed | PM$_{10}$ (µg m$^{-3}$) average (RSD) |
|---|---|---|---|
| BUD (Hungary) | 21.01–10.02.2019 | 10 | 58.6 (18) |
| | 01.07–07.07.2019 | 7 | 22.9 (17) |
| | 07.01–29.01.2020 | 21 | 47.4 (35) |
| KPS (Hungary) | 21.01–10.02.2019 | 10 | 41.1 (11) |
| | 08.07–14.07.2019 | 7 | 10.0 (23) |
| | 07.01–27.01.2020 | 21 | 39.0 (36) |
| MSK (Hungary) | 14.01–03.02.2019 | 10 | 64.6 (20) |
| | 24.06–30.06.2019 | 7 | 23.5 (23) |
| | 07.01–27.01.2020 | 21 | 53.6 (30) |
| PUT (Hungary) | 14.01–03.02.2019 | 10 | 81.0 (18) |
| | 24.06–30.06.2019 | 7 | 17.9 (26) |
| | 07.01–27.01.2020 | 21 | 55.8 (43) |
| VES (Hungary) | 28.01–17.02.2019 | 10 | 34.8 (23) |
| | 08.07–14.07.2019 | 7 | 10.5 (17) |
| | 07.01–27.01.2020 | 21 | 32.8 (32) |
| BUC-R (Romania) | 22.01-11.02.2019 | 10 | 52.1 (17) |
| | 19.06-25.06.2019 | 7 | 32.4 (11) |
| | 06.02-26.02.2020 | 19 | 36.5 (46) |
| BUC-M (Romania) | - | - | - |
| | - | - | - |
| | 06.02-26.02.2020 | 21 | 34.4 (53) |
| CLJ (Romania) | 26.01-15.02.2019 | 10 | 52.9 (22) |
| | 19.06-25.06.2019 | 7 | 17.2 (26) |
| | 10.01-03.02.2020 | 21 | 39.9 (62) |
| DEV (Romania) | 30.01-19.02.2019 | 10 | 67.8 (16) |
| | 02.07-08.07.2019 | 6 | 28.1 (57) |
| | 08.01-28.01.2020 | 21 | 71.3 (34) |
| FOC (Romania) | 19.02-11.03.2019 | 10 | 61.2 (25) |
| | 19.06-25.06.2019 | 7 | 28.0 (12) |
| | 09.01-29.01.2020 | 20 | 49.5 (46) |
| | | Total: 359 | |

**Table 1. The sampling dates, the number of samples analysed and the average PM$_{10}$ concentrations with relative standard deviations (RSD).**

115



Among the samples collected in the winter of 2019 those with the 10 highest $PM_{10}$ concentrations were selected for GC-MS analysis, whereas from 2021 nearly all samples collected on consecutive days were analysed. This is also reflected in the $PM_{10}$ mass concentrations reported in Table 1, as the average $PM_{10}$ concentrations are typically higher for the winter of 2019 than for 2020. During the sampling periods blank samples were collected at each sampling site and were also analysed.

**2.2 GC-MS analysis of the samples for tracer compounds**

The GC-MS method for the analysis of tracer compounds was adopted from Hoffer et al. (2021). From the ambient samples 18.1 $cm^2$ of the filters were extracted 3 times in dichloromethane-methanol 2:1 mixture. The extraction efficiency was followed by a recovery standard (p-terphenyl-$d_{14}$) added to the samples before the extraction. During the first extraction step 7.5 ml of dichloromethane : methanol (2:1) mixture was applied to the filter portion and the sample was shaken in a vortex agitator for 1 h. The volume of the second and third extracts was 6 and 5 ml, respectively. The combined extracts were then filtered through a syringe filter (0.45 μm) and dried under $N_2$ stream. The redissolved (in dichloromethane : methanol 2:1 mixture) sample extracts were then directly analysed for the less polar tracer compounds, while the more polar compounds (levoglucosan, terephthalic acid, melamine) were measured after derivatisation with BSTFA-TMCS (N,O-Bis(trimethylsilyl)trifluoroacetamide-Trimethylchlorosilane, 99:1, (Sigma-Aldrich) and pyridine (anhydrous, Merck) (1:1 volume ratio) at 80 °C for 1 h, with an Agilent 6890 gas chromatograph coupled to a Agilent 6973 mass spectrometer. The column type and the temperature programs were the same as in Hoffer et al. (2021). In the samples the amount of m-terphenyl (m-TPH), p-terphenyl (p-TPH), the 1,3,5-triphenylbenzene (135-TPB), the 1,2,4-triphenylbenzene (124-TPB), the quaterphenyl isomers (m,p-QTPH, p-QTPH), 2-(benzoyloxy)ethyl vinyl terephthalate (2-BEVT), 5-hexene-1,3,5-triyltribenzene (styrene trimer, SSS), the 2-methylene-4,6-diphenylhexanenitrile (ASS), 2-phenethyl-4-phenylpent-4-enenitrile (SAS), 4,6-diphenylhept-6-enenitrile (SSA), levoglucosan (LGS), melamine and terephthalic acid (TPA) were measured. The identification of these compounds was based on the method reported in Hoffer et al. (2021). The calibration of the instrument was performed with standard solutions, but for the lack of available individual standards the mass concentrations of some compounds were quantified based on calibration curves of other similar compounds: the terphenyls were expressed as p-TPH, 124-TPB and quaterphenyls as well as the 2-BEVT as 135-TPB, the ASS, SSA and SAS as SSS as in detailed in Hoffer et al. (2021).

Blank samples were also analysed and the LOQ values were calculated as the average of the blanks + 10 times the standard deviation of the blanks. The LOQ values were obtained separately for the 2019 and 2020 winter campaigns as well as for the summer campaign separately for the Hungarian and Romanian samples.

**3 Results and discussion**

**3.1 Burning of waste containing PET**

The tracers for the emissions from the burning of PET-containing wastes (usually PET bottles and textiles containing "polyester") can be manifold, but recently we have identified a highly specific 2-(benzoyloxy) ethyl vinyl terephthalate (2-BEVT) (Hoffer et al., 2021). Figure 2. shows the average 2-BEVT-to-levoglucosan (LGS) ratio obtained at the different sampling sites in Hungary and Romania for the winter samples. These mass



concentration ratios may be used to weigh the relative abundance of PET burning emissions to firewood burning among the different settlements. The mass concentrations of 2-BEVT were in some samples below the

quantification limit (LOQ), these values were considered zero in calculating the average. The numbers shown inside the bars represent the percentage of cases in which the 2-BEVT were determined quantitatively (the data coverage of the LGS concentrations were 100%), the error bars show the standard deviation of the data.

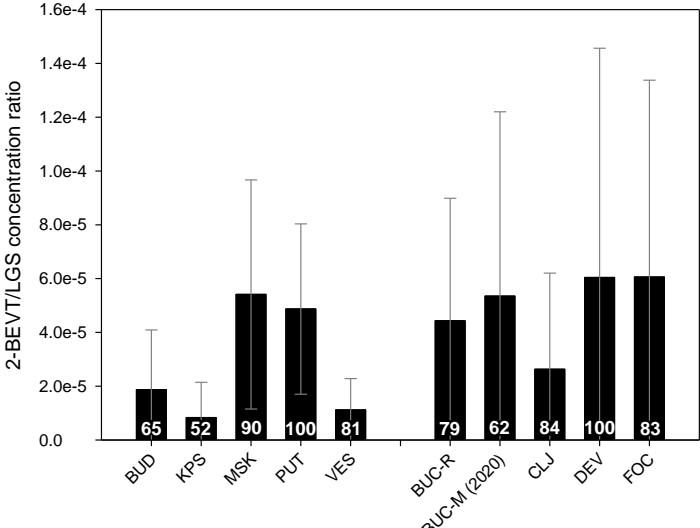

**Figure 2. The average mass concentration ratios of 2-BEVT to LGS in the PM$_{10}$ samples collected at the different**
**locations in the winters of 2019 and 2020. The numbers shown inside the bars represent the percentage of cases in which the concentrations of 2-BEVT were above the limit of quantification. The error bars represent the standard deviation of all data.**

The figure shows that in Hungary the relative contribution of PET burning is significantly higher in the north-
eastern region (MSK and PUT) than elsewhere in the country. The difference may be up to a factor 5 between the most and the least affected settlements (MSK and VES). It should be noted that in terms of wintertime PM$_{10}$ concentrations (see Table 1) the trends are identical, but the concentration ratios are much lower (only about a factor of 2 between MSK and VES). In Putnok 2-BEVT was quantified in all analysed samples, whereas its concentration was above the limit of quantification only in half of the samples collected at the background
station of K-puszta. Since the mass concentrations of levoglucosan were the highest in Putnok (Figure S1), the combination of these results yields that the highest absolute 2-BEVT concentrations were also measured there: on average, the mass concentrations of 2-BEVT found in Putnok was about a factor of 5 higher than that in Budapest, and up to a factor of 15 (!) higher than that found in Veszprém. These findings are by far out of proportion to the differences in ambient PM$_{10}$ concentrations. It should be noted that since lignite is a frequently
used solid fuel in Putnok and the emission factor of levoglucosan from lignite combustion may be higher than in the case of wood burning (Fabbri et al., 2009), this may lead to some bias in the observed 2-BEVT to levoglucosan ratios. In Veszprém, the relative importance of PET burning is found to be much lower even



though its tracer was quantified in the vast majority (80%) of the $PM_{10}$ samples. On the contrary, 2-BEVT was quantified only in 66% of the $PM_{10}$ samples in Budapest, yet the relative share of PET burning was found to be higher on average than that in Veszprém.

In the case of the $PM_{10}$ samples collected in Romania, the relative share of PET burning on average was found to be largely comparable with those obtained at the most polluted Hungarian sites, with the exception of Cluj-Napoca. At the sampling locations in Bucharest (Măgurele) the 2-BEVT was quantified in about 70% of the $PM_{10}$ samples and its concentration ratio to LGS was among the highest. This implies quite intensive PET-containing waste burning in the region. According to the measured 2-BEVT concentrations the sites most heavily impacted by PET burning were Deva and Focsani in the study period. In Deva 2-BEVT was quantified in all $PM_{10}$ samples, the corresponding ratios in Focsani and Cluj-Napoca were 83% and 84%, respectively. However, in the latter city the relative share of PET burning was found to be relatively low (but higher than that obtained for Budapest). Although the numbers of the analysed samples were different in the two heating seasons, the concentration ratios of 2-BEVT to LGS at the majority of the sampling locations were largely comparable in the two years, with the exception of Cluj-Napoca and Focsani at which waste burning was more intense in 2020 than in 2019. In the summertime samples the concentrations of 2-BEVT were below the limit of quantification in all but one of the samples.

It has been found recently that the burning of PET and PET-containing textiles releases significant amounts of m-terphenyl and p-terphenyl (Hoffer et al., 2021). While these compounds are not reported in particulates emitted by biomass burning and fossil fuel combustion, they were identified in lignite smoke (Fabbri et al., 2009). It has been shown that besides the burning of PET-containing wastes the combustion of PS, ABS, and paper (the latter is for the meta isomer) are also significant sources of terphenyls. In addition, the relative emission factors of the meta- and para- isomers are different for different types of wastes. It was observed that p-terphenyl is emitted in higher quantities than m-terphenyl in the burning of PET-containing waste (e.g. PET flasks and clothes containing polyester), whereas for other waste types the ratio of the emission factor of the para- to meta-terphenyl was lower than 1 (0.6 and 0.4 for PS and ABS, respectively). Figure 3 shows the average ratios of the mass concentrations of m-terphenyl and p-terphenyl at the different sampling sites.

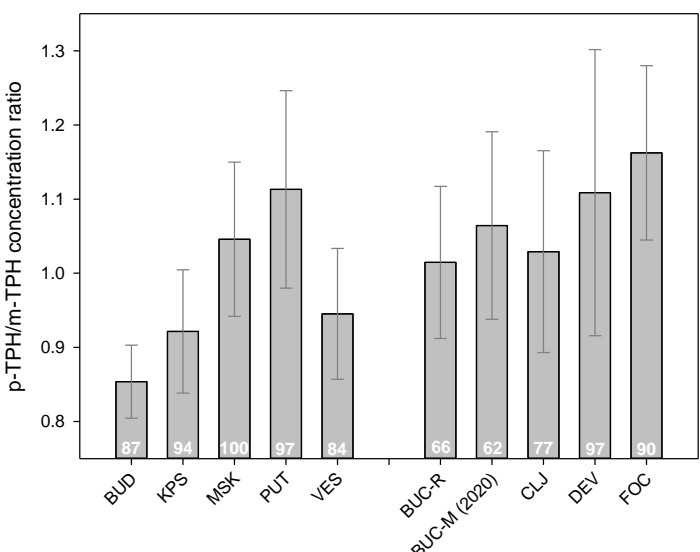

**Figure 3. The average mass concentration ratio of p-TPH and m-TPH in the PM$_{10}$ samples collected at the different sampling sites. The numbers shown inside the bars represent the percentage of cases in which the concentrations of both isomers were above the limit of quantification. The error bars represent the standard deviation of all data.**

In spite of the fact that there is no statistically significant correlation between the 2-BEVT/LGS and the p-TPH/m-TPH concentration ratios obtained for the PM$_{10}$ samples collected at a given sampling site, it can be clearly seen in Figure 2. that at the sampling locations implying higher incidence of PET-waste burning (MSK, PUT, BUC-M, DEV, FOC) the ratios of the two terphenyl isomers are all higher than 1. Although there might be other emission sources of the terphenyl isomers (such as lignite burning), these findings also confirm that PET burning may indeed be a non-negligible emission source of particulates in the above settlements. This conclusion is also supported by the very high emission factor of the terphenyls from PET burning (Hoffer et al., 2021), but obviously, since the p-terphenly to m-terphenyl ratio is lower than that typical for PET burning emissions (1.1–1.7), the contribution of other sources to the terphenyl concentration should also be considered.

### 3.2 Burning of fibreboard, PS and paper

The styrene trimer (SSS) is emitted mainly during the incomplete burning of polystyrene, but its relative emission factor (ng mg$^{-1}$ PM$_{10}$$^{-1}$) was also considerable in the case of LDF burning (coated low-density fibreboard used as furniture material), as well as during paper burning (printed and coated, waxy paper burning) (Hoffer et al., 2021). On a *mass* basis about 2 orders of magnitude more SSS is emitted from PS than from LDF or PAP, but since the latter two are burned in much more significant quantities than PS their contributions to its atmospheric concentration are at least comparable. Figure 4 shows the concentration ratio of the styrene trimer to the LGS measured in the ambient PM$_{10}$ samples collected in Hungary and Romania.

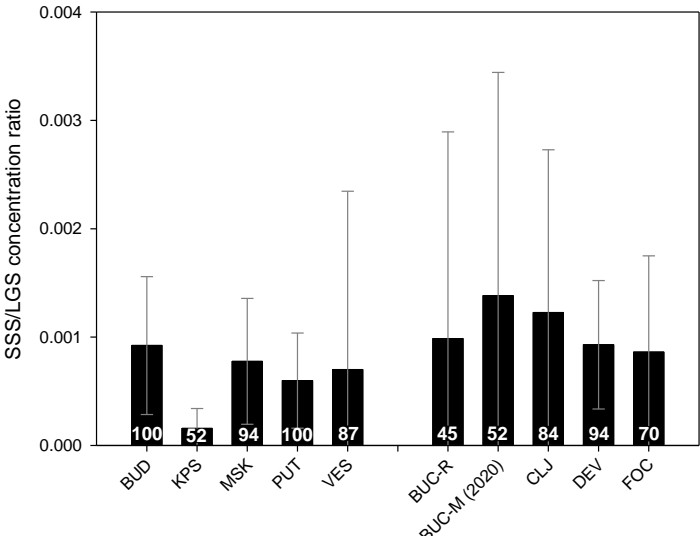

**Figure 4. The average relative mass concentration ratio of styrene trimer (SSS) to levoglucosan in the PM$_{10}$ samples at the different sampling sites. The numbers shown inside the bars represent the percentage of cases in which the concentrations of styrene trimer (SSS) were above the limit of quantification. The error bars represent the standard deviation of all data.**

The average relative concentration of SSS was the lowest in the Hungarian background station KPS, and SSS were quantified in only half of the samples collected at this station. On the contrary, SSS was measured in all PM$_{10}$ samples from Budapest and Putnok, and in most of the samples collected in Miskolc and Veszprém. The concentration ratios of SSS/LGS were within a factor of two at all sampling sites in Hungary except KPS. The average SSS/LGS ratio was slightly higher in most of the PM$_{10}$ samples in Romania (except FOC). In the latter town there was a large year-to-year variability of the SSS concentration, as in 2019 its concentration was above the LOQ in only 20% of the samples, whereas in 2020 the same ratio was 95%. It is important to note that in the samples collected near Bucharest SSS was identified only in about half of the samples (in BUC-R 60% and 37% in 2019 and 2020, respectively), but the average SSS/LGS ratio was among the highest, which may indicate that waste burning may either be highly episodic or is related to some specific wind sectors only. It is interesting to see that the SSS/LGS ratio is higher in towns with larger population than in smaller settlements (BUD>MSK>VES and BUC, CLJ>DEV, FOC). This might indicate that the types of waste yielding SSS upon burning are more readily accessible in larger cities per unit number of wood-burning households. The concentrations of the SSS were below the LOQ in the PM$_{10}$ samples collected during summer in both countries.

Hoffer et al. (2021) identified 135-TPB in most of the PM$_{10}$ samples collected during the burning of different types of solid waste in the laboratory. This compound has been considered as a universal tracer for waste burning (Simoneit et al., 2005). 1,3,5-triphenylbenzene was quantified in almost all (> 93%) PM$_{10}$ samples during winter. In the study period the average atmospheric concentrations of 135-TPB were the highest in Bucharest and Cluj-Napoca (1.3 ng m$^{-3}$ and 1.6 ng m$^{-3}$, respectively). These concentrations are at the lower end of those obtained for Beijing during the summer season of 2008 (1.58–4.58 ng m$^{-3}$, Li and Fang 2009), but were markedly higher than





that measured in South Poland in the heating season of 2017 (0.8 ng m$^{-3}$, Furman et al., 2020). In Hungary the average atmospheric concentrations of the 135-TPB varied between 0.25 and 0.42 ng m$^{-3}$ at the different sampling stations.

Unlike the 2-BEVT and SSS, the relative concentration of the compounds containing 4 aromatic rings (the quaterphenyls (m,p-QTPH, p-QTPH) and the triphenyl-benzene isomers (135-TPB and 124-TPB) also show relatively high relative concentration values (see Figure 5) at the background station (KPS) indicating that these compounds likely have longer residence time in the atmosphere (as follows from the absence of the double bond which is present both in the 2-BEVT and SSS). In Hungary the relative concentrations of quaterphenyls to LGS

were the highest in MSK where PET burning was also found to be significant. Similarly in Romania this ratio was the highest in Bucharest where the relative concentrations of both the SSS and the 2-BEVT were also the largest. Although the relative concentrations of the quaterphenyls to LGS are highly variable at different stations their ratio is only slightly higher (by 20–50%) in the PM$_{10}$ samples collected in Romanian settlements than those in Hungary. On the other hand, the relative amount of the triphenylbenzenes to the LGS show much larger differences between the two countries, their relative concentrations are larger in the PM$_{10}$ samples collected in

differences between the two countries, their relative concentrations are larger in the PM$_{10}$ samples collected in Romania on average by a factor of 2–3 than those in Hungary. Exceptionally high concentrations of the triphenylbenzenes relative to LGS were obtained near Bucharest and in Cluj-Napoca. It is important to note that the large sample-to-sample variability (the large standard deviation of the measured ratio, see Figure 5) at these stations may indicate the presence of strong and intermittent emission source(s) in the vicinity of the sampling

sites. Li and Fang speculated that in Beijing the emission sources of 135-TPB might also be waste incineration plants and fossil fuel combustion, in that case 135-TPB may not be a reliable tracer for household waste burning. On the other hand, if the large variation in the atmospheric concentration of triphenylbenzenes can be attributed to the substantial changes in the types of solid waste that is burned (e.g. the emission factor of triphenylbenzenes for PS burning is very large), it may still be a generic tracer of waste burning.

Unlike other more specific waste burning tracers both quaterphenyls and triphenylbenzenes were also quantified in PM$_{10}$ samples collected during summer (Figure S2). This might imply that there are other industrial or traffic-related emission sources. Furman et al. (2020) found that outside the heating season in south Poland the relative concentration of 135-TPB to PM$_{10}$ was still about 60% of that found during the winter. Here we also note that the p-quaterphenyl is emitted from aircraft engines, and it was suggested as a tracer for PAH pollution from

airplanes (Krahl et al., 1998).

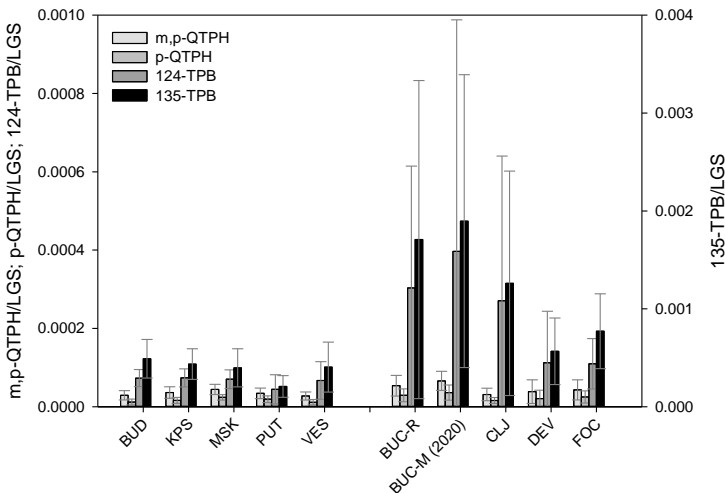

**Figure 5. The average relative mass concentration ratio of quaterphenyls and triphenylbenzenes to levoglucosan in the PM$_{10}$ samples at the different sampling sites. The error bars represent the standard deviation of all data. The percentage of cases in which the concentrations of the presented compounds were above the limit of quantification was larger than 87%.**


The ratio of the triphenylbenzenes and quaterphenyls showed large variation during the burning of different plastic types in the laboratory, the concentration of the triphenylbenzenes was much larger during the burning of waste specimens containing styrene and/or also emitting SSS (PS, LDF, PAP, ABS), than in the case of PET and

RAG burning (Hoffer et al., 2021). The average m,p-QTPH/124-TPB ratios were <0.3 and 1.3–2.3 for the styrene containing wastes (LDF, ABS, PS, PAP) and for the PET-containing wastes, respectively.

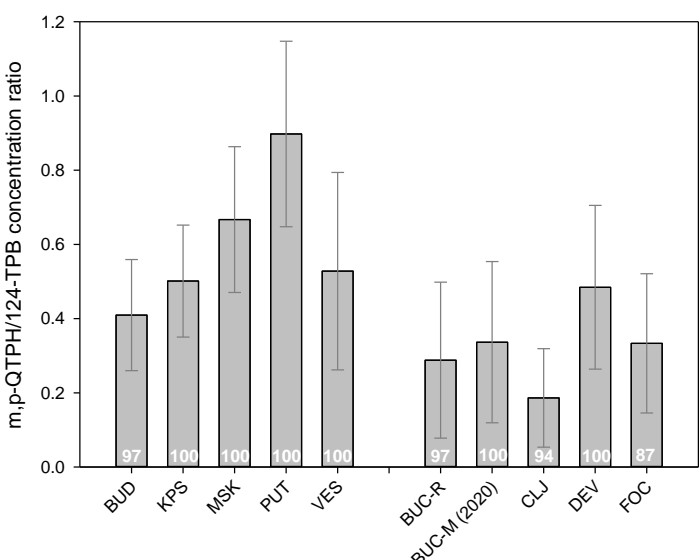

**Figure 6. The average relative concentration ratio of m,p-QTPH to 124-TPH at the different sampling locations. The numbers shown inside the bars represent the percentage of cases in which the concentrations of both isomers were**
**above the limit of quantification. The error bars represent the standard deviation of all data.**

Figure 6 shows that this ratio is the largest in MSK and PUT, implying that large amount of PET containing wastes is burned in the region. In Romania lower m,p-QTPH/124-TPB ratios were found at all stations, which, together with the elevated concentrations of SSS, suggested that large quantities of styrene containing waste

(PS/LDF/PAP) were burned in the vicinity of the sampling sites. The higher m,p-QTPH/124-TPB concentration ratios found at DEV and FOC are supporting enhanced rate of PET burning (2-BEVT/LGS ratio) at these locations. It should be noted that in Hungary and Romania the concentration ratio of m,p-QTPH/124-TPB is below 1, which implies that the concentration of these components might be determined by the burning of LDF/PS/PAP in both countries.

While SSS can be emitted from various sources melamine is more specific for the burning of materials containing melamine-formaldehyde resins e.g. coated fibreboards used as furniture parts. Based on the melamine/LGS ratio (Figure 7) the extent of the burning of furniture panels can be compared at the different sampling sites. Here we note that melamine concentrations were below the LOQ (except one sample from Putnok) in the $PM_{10}$ samples collected in the winter of 2019 in Hungary, and were quantified only about half

(43–57%) of the samples from 2020. In the $PM_{10}$ samples collected in Romania this compound was quantified in the majority (50–100%) of the samples in both years, indicating that the burning of furniture panels is likely more common in Romania than in Hungary (Figure 7). It should also be noted that in both countries the burning of furniture panels is more common in the capitals than in other settlements, possibly due to the much wider availability of scrap furniture in the high-income regions.

Melamine was also sporadically identified in some Hungarian and Romanian samples collected during summer, but its relative contribution to $PM_{10}$ concentrations was much lower than during the winter. There might be some



sporadic burning of furniture panels in summer or emissions from local furniture plants (e.g. manufacturing and tailoring of furniture panels).

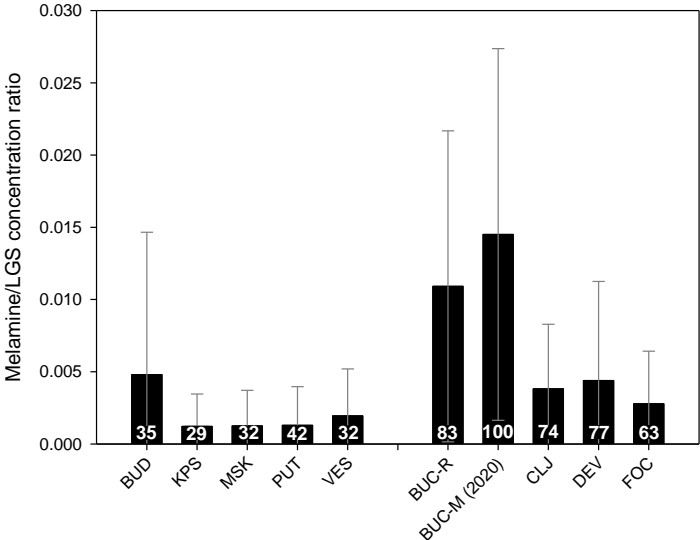

**Figure 7. The relative concentration of melamine to levoglucosan in PM$_{10}$ samples from the heating periods. The numbers shown inside the bars represent the percentage of cases in which the concentrations of melamine were above the limit of quantification. The error bars represent the standard deviation of all data.**

In the samples collected during the heating season the pyrolysis products of the ABS (ASS, SSA, SAS) were also identified (Figure 8). These compounds are released into the atmosphere not only from the burning of scrap electronic devices, but also upon the burning of furniture panels as it is widely used as edge banding tapes for chipboards, MDF, or HDF. Similarly to melamine, the relative concentrations of the different ABS pyrolysis products to the LGS were the highest in the samples collected in Budapest and Bucharest also supporting that scrap electronic devices and furniture are more readily available for burning in large cities.



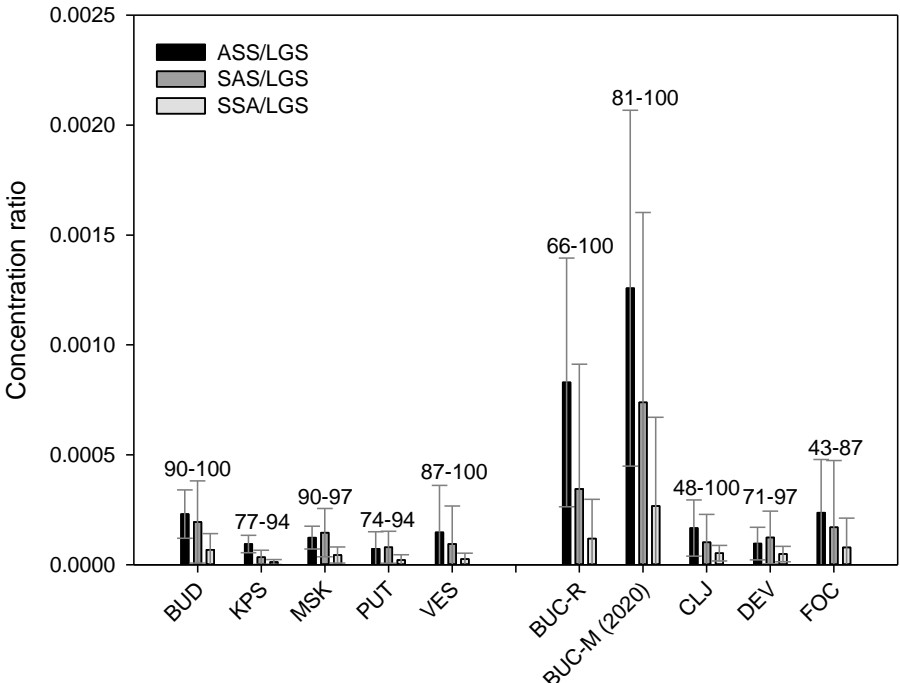


**Figure 8. The concentration ratio of ABS pyrolysis products to LGS in PM$_{10}$ samples from the heating seasons. The error bars represent the standard deviation of all data, the numbers above the error bars indicate the frequency of occurrence expressed in percentages.**

**3.3 Assessment of potential contribution of household waste burning to ambient PM$_{10}$ mass concentrations**

The contribution of particulate emissions from household waste burning to ambient PM$_{10}$ mass concentrations in the Hungarian and Romanian sampling sites was assessed for the first time incrementally from the measured atmospheric concentrations of quaterphenyls, 2-BEVT, SSS, ASS, SAS, SSA and melamine using estimated waste component-specific EFs based on the data reported by Hoffer et al. (2021). The component-specific EFs were weighted averages for waste mixtures based on their estimated abundance in household waste, as some

compounds are emitted from multiple sources. For this estimation furniture panel, paper, rag, and household waste containing a mixture of different plastics were treated separately. Taking into account the plastic composition of municipal waste (Bodzay and Bánhegyi, 2016), it was assumed that the plastic-type household waste consisted of 42% PE, 28% PET, 14% PP and PS, 0.7% PVC and ABS. Furthermore, EFs were calculated for waste mixtures in which the mass of furniture boards varied between 10 and 91.7% and that of the rag, paper

and mixed plastics from household waste between 1.8% and 65.2%. Table 2 shows the relative EFs of specific tracers as well as the absolute PM$_{10}$ emission factors for the burning of individual waste types (taken from Hoffer et al., 2020), and summarises the range of EFs as well as the average for the prescribed real-life waste mixtures whose compositional ranges are shown in the first column of the table.





| Waste type | m/m % in mixture | PM$_{10}$ EF (relative scale) | Relative EF (µg g$^{-1}$ PM$_{10}^{-1}$) | | | | | | | | | | | |
|---|---|---|---|---|---|---|---|---|---|---|---|---|---|---|
| | | | 135-TPB | 124-TPB | m-TPH | p-TPH | m,p-QTPH | p-QTPH | 2-BEVT | SSS | ASS | SAS | SSA | Melamine |
| ABS | 0.013 - 0.46 | 38 | 24 | 10 | 250 | 100 | 3.2 | 0 | 0 | 0 | 22 | 43 | 15 | 0 |
| LDF | 10 - 92 | 2 | 23 | 2.2 | 11 | 0 | 0 | 0 | 0 | 380 | 200 | 350 | 180 | 19000 |
| PAP | 1.8 - 65 | 1 | 75 | 10 | 290 | 0 | 2.9 | 0 | 0 | 380 | 51 | 18 | 17 | 0 |
| PE | 0.78 - 28 | 9 | 0 | 0.43 | 17 | 0 | 0 | 0 | 0 | 0 | 0 | 0 | 0 | 0 |
| PET | 0.52 - 18 | 5 | 100 | 390 | 5400 | 7700 | 920 | 590 | 250 | 0 | 0 | 0 | 0 | 0 |
| PP | 0.26 - 9.2 | 16 | 0.49 | 0.21 | 57 | 0 | 0 | 0 | 0 | 0 | 0 | 0 | 0 | 0 |
| PS | 0.26 - 9.2 | 25 | 800 | 190 | 1100 | 690 | 34 | 9.7 | 0 | 2900 | 0 | 0 | 0 | 0 |
| PVC | 0.013 - 0.46 | 17 | 6.2 | 0 | 67 | 24 | 1.9 | 0 | 0 | 0 | 0 | 0 | 0 | 0 |
| RAG | 1.8 - 65 | 4 | 15 | 78 | 610 | 920 | 100 | 67 | 6.9 | 0 | 0 | 0 | 0 | 0 |
| Mixture minimum | | | 45 | 19 | 183 | 1106 | 96 | 79 | 15 | 491 | 98 | 136 | 73 | 19000 |
| Mixture maximum | | | 343 | 101 | 959 | 2392 | 250 | 167 | 196 | 2449 | 197 | 344 | 177 | 19000 |
| Mixture average | | | 159 | 64 | 593 | 1783 | 165 | 125 | 89 | 1248 | 157 | 260 | 135 | 19000 |

**Table 2. The estimated emission factors for different tracer compounds for the burning of waste mixtures.**

It should be noted that the assessment of the contribution of particulate emission from residential waste burning to the ambient PM$_{10}$ concentration are loaded with very high uncertainty emerging from the large variations of the EFs (which depend on the burning conditions and the type of waste burned), and also from the fact that most tracer compounds are emitted upon the burning of several waste types (e.g. SSS from PS but also from paper and LDF) with vastly different emission factors. In addition, all waste burning tracers are considered refractory in the atmosphere, i.e. they do not decompose during atmospheric transport. Because of all of these inherent uncertainties the assessment should only be treated as a first order approximation. Table 3 summarises the estimated contributions of waste burning particulate emissions to ambient PM$_{10}$ concentrations based on the measured atmospheric concentrations of the different tracer compounds as well as their relative EFs taken from laboratory measurements.

| Tracer component: | 135-TPB | 124-TPB | m-TPH | p-TPH | m,p-QTP | p-QTPH | 2-BEVT | SSS | ASS | SAS | SSA | Melamine |
|---|---|---|---|---|---|---|---|---|---|---|---|---|
| Most characteristic waste type(s): | PS/LDF/ PET/PAP | PET/PS | PET/PS/ PAP | PET/PS | PET/PS | PET/PS | PET | PS/LDF/ PAP | LDF/PAP/ ABS | LDF/PAP/ ABS | LDF/PAP/ ABS | LDF |
| EF (µg g$^{-1}$ PM10$^{-1}$): | 200 | 100 | 600 | 1800 | 200 | 100 | 100 | 1200 | 200 | 300 | 100 | 19000 |
| KPS | + | 1.1 | 0.3 | 0.1 | 0.3 | 0.3 | 0.1 | 0.2 | 0.7 | 0.2 | 0.2 | 0.1 |
| VES | + | 1.3 | 0.3 | 0.1 | 0.3 | 0.2 | 0.2 | 1.1 | 1.6 | 0.7 | 0.5 | 0.2 |
| BUD | + | 1.2 | 0.4 | 0.2 | 0.3 | 0.2 | 0.4 | 1.4 | 2.0 | 1.2 | 1.3 | 0.5 |
| MSK | + | 1.4 | 0.5 | 0.2 | 0.4 | 0.5 | 1.2 | 1.4 | 1.2 | 1.0 | 0.9 | 0.1 |
| PUT | + | 1.3 | 0.6 | 0.2 | 0.5 | 0.6 | 1.5 | 1.5 | 1.1 | 0.8 | 0.7 | 0.2 |
| CLJ | +++ | 7.9 | 0.2 | 0.1 | 0.4 | 0.4 | 0.7 | 2.4 | 1.8 | 0.8 | 1.2 | 0.5 |
| DEV | ++ | 2.9 | 0.3 | 0.1 | 0.4 | 0.5 | 1.4 | 2.0 | 1.2 | 1.0 | 1.2 | 0.7 |
| FOC | ++ | 2.1 | 0.3 | 0.1 | 0.4 | 0.5 | 1.3 | 1.5 | 2.3 | 1.2 | 1.6 | 0.3 |
| BUC-R | +++ | 5.0 | 0.3 | 0.1 | 0.5 | 0.5 | 0.8 | 1.4 | 7.1 | 2.3 | 2.3 | 1.0 |
| BUC-M (2020) | +++ | 7.1 | 0.3 | 0.1 | 0.6 | 0.6 | 1.0 | 2.1 | 10.0 | 4.3 | 4.6 | 1.2 |

**Table 3. Estimated mass contributions of waste burning particulate emissions to ambient PM$_{10}$ concentrations at the different sampling sites in Hungary and Romania based on the measured atmospheric concentrations of the different tracer compounds as well as their relative EFs taken from laboratory measurements. The data are in percentages (m/m) of the measured PM$_{10}$ concentrations.**

Table 3 demonstrates that the estimated contribution of PM$_{10}$ emitted from the burning of different waste types in households to atmospheric PM$_{10}$ concentration is in the order of a few per cent based on the measured ambient concentrations and relative emission factors of waste burning tracers. Since there may be additional sources of 135-TPB (its relative contributions to ambient PM$_{10}$ were also significant in summer at some sampling sites), and the very large uncertainty in its EF, in the column based on its ambient concentrations only the magnitude of the estimated contribution of waste burning is indicated in the table. Although the data calculated from the individual tracers should not be added due to the considerable overlap between the sources, PET/PS and



LDF/PAP/ABS-related set of data may, since they represent distinctly different types of wastes burned. By doing so the estimated contributions of waste burning particulate emissions to ambient $PM_{10}$ were found to be lowest for the regional background station KPS and for Veszprém. In Budapest there is a strong indication that scrap furniture was burned in significant quantities in household stoves and possibly in repair shops, whereas in MSK and PUT the burning of common household waste (PET and/or RAG) was more typical. The results also imply that in Bucharest and Cluj the burning of scrap furniture was also quite common, but accompanied with higher contributions from PET and other household waste burning compared to those found for Budapest. In Deva and Focsani the contribution of PET and/or RAG burning emissions was predominant similarly to that found in PUT and MSK, with some additional contributions from the burning of other waste types (e.g. PS, LDF).

### 3.4. Assessment of the quantities of household waste burned in households and repair shops

Using the absolute emission factors of the different tracer compounds determined upon controlled waste burning in the laboratory as well as reported emission factors for levoglucosan from wood burning the quantities of solid wastes burned in households may be estimated relative to the amount of firewood for which statistical data are available. This assessment can only be considered as back-of-the-envelope calculations because of the vast uncertainties associated with emission measurements, variable burning conditions, degree of co-firing, atmospheric stability of tracers, as well as all other underlying assumptions. Table 3 summarises the obtained results along with the applied EFs which were estimated similarly as in the case of the relative EFs (Table 2) using the reported data from Hoffer et al. (2021). For the calculations we assumed that levoglucosan is emitted solely from wood burning at a rate of 200 mg kg$^{-1}$ (Jimenez et al., 2017).

| Tracer component: | 135-TPB | 124-TPB | m-TPH | p-TPH | m,p-QTP | p-QTPH | 2-BEVT | SSS | ASS | SAS | SSA | Melamine |
|---|---|---|---|---|---|---|---|---|---|---|---|---|
| Most charasteristic waste type(s): | PS/LDF/ PET/PAP | PET/PS | PET/PS/ PAP | PET/PS | PET/PS | PET/PS | PET | PS/LDF/ PAP | LDF/PAP/ ABS | LDF/PAP/ ABS | LDF/PAP/ ABS | LDF |
| EF (mg kg$^{-1}$): | 1.4 | 0.62 | 5.90 | 30.00 | 1.9 | 2.1 | 0.9 | 9.9 | 0.5 | 0.85 | 0.43 | 51 |
| KPS | + | 2.4 | 0.4 | 0.07 | 0.4 | 0.2 | 0.2 | 0.3 | 3.8 | 0.8 | 0.6 | 0.5 |
| PUT | + | 1.4 | 0.4 | 0.09 | 0.4 | 0.2 | 1.1 | 1.2 | 2.9 | 1.9 | 1.0 | 0.5 |
| VES | + | 2.2 | 0.3 | 0.06 | 0.3 | 0.1 | 0.3 | 1.4 | 5.9 | 2.2 | 1.2 | 0.8 |
| DEV | + | 3.6 | 0.3 | 0.06 | 0.4 | 0.2 | 1.3 | 1.9 | 3.8 | 2.9 | 2.3 | 1.7 |
| MSK | + | 2.3 | 0.5 | 0.11 | 0.5 | 0.2 | 1.2 | 1.6 | 4.9 | 3.4 | 2.1 | 0.5 |
| CLJ | ++ | 8.7 | 0.2 | 0.04 | 0.3 | 0.2 | 0.6 | 2.5 | 6.7 | 2.4 | 2.4 | 1.5 |
| BUD | + | 2.4 | 0.5 | 0.11 | 0.3 | 0.1 | 0.4 | 1.9 | 9.2 | 4.6 | 3.2 | 1.9 |
| FOC | ++ | 3.5 | 0.3 | 0.08 | 0.5 | 0.2 | 1.3 | 1.7 | 9.5 | 4.0 | 3.7 | 1.1 |
| BUC-R | +++ | 9.8 | 0.3 | 0.06 | 0.6 | 0.3 | 1.0 | 2.0 | 33 | 8.1 | 5.5 | 4.3 |
| BUC-M (2020) | +++ | 13 | 0.3 | 0.07 | 0.7 | 0.3 | 1.2 | 2.8 | 50 | 17 | 12 | 5.7 |

**Table 4. The estimated mass of solid waste burned relative to firewood (in percentage) at the different locations based on the measured ambient concentration of different tracer compounds. The first row shows the absolute emission factors of waste burning tracers as determined experimentally in Hoffer et al. 2021.**

The results summarised in Table 4 indicate that the mass of the household waste burned relative to that of firewood is in the order of a few percent (up to 5%), but at some sites (BUD, BUC, DEV, FOC) in Romania it is somewhat larger, primarily as a consequence of excess LDF/PAP/ABS burning. Note that different tracers used for the same types of wastes burned yield largely similar order of magnitudes in the estimations. It can be seen that on a mass basis LDF/PAP/ABS dominate the types of waste burned, most likely from the burning of scrap furniture. By assuming that on a mass basis the burned waste is 3–5% of that of the firewood in both countries and taking into account that the total mass of firewood annually consumed in Hungary and Romania in the year



2019 were 7.4 and 12.2 million tonnes, respectively (Clean Air Action Group, Romanian Statistical Office), the
calculations reveal that 8–13 % and 9–15 % of all household waste produced in Hungary and Romania,
respectively, end up being burned in household stoves. (According to EUROSTAT
(https://ec.europa.eu/eurostat/databrowser/view/ten00106/default/table?lang=en) data the amounts of household
wastes produced in Hungary and Romania were 2.7 and 4.1 million tonnes year$^{-1}$, respectively.) These are
alarming number given that such activities are prohibited in both countries and all over the EU.

**4 Summary**

The tracer compounds identified by Hoffer et al. (2021) for the burning of different waste types were quantified
in atmospheric $PM_{10}$ samples collected in different settlements in Hungary and Romania in two consecutive
heating seasons of 2019/2020 and 2020/2021. As a reference $PM_{10}$ samples were also collected in summer at all
locations. In spite of the fact that some tracers contain double bonds and therefore their long-term atmospheric
stability is questionable, they were quantified in $PM_{10}$ samples collected in winter, but were missing from the
summer samples. The relative contributions of PET burning emissions to that of wood burning were followed by
the concentration ratio of 2-BEVT and LGS. The results showed that in Hungary large quantities of PET-
containing wastes are burned in the north-eastern part of the country. Although they can be emitted from other
sources as well, the burning of PET was supported by the higher concentration ratios of the p-terphenyl to m-
terphenyl isomers at those sites. Implications for the excessive burning of PET were also found in Romanian
cities where the burning of PET-containing waste was found to be on a similar scale than in the most polluted
north-eastern regions of Hungary. The styrene trimer (SSS) can be used as a tracer for the burning of furniture
panels, household wastes containing PS, and printed advertising and waxy papers, whereas melamine was used
as a tracer for the burning of furniture panels and/or wastes containing melamine-formaldehyde resin. Larger
SSS emissions were observed in the capitals possibly due to the burning of large quantities of scrap furniture
panels as well as other types of styrene-containing household waste (such as printed and waxy papers of
advertising leaflets). The presence of ASS, SSA and SAS also implied the burning of furniture panels, as these
compounds are emitted from the burning of furniture panel edge bands. The presence of quaterphenyls and
triphenylbenzenes indicated the burning of PET, RAG, and to a lesser extent PS, ABS, and furniture panels.
The estimated contributions of waste burning emissions to the mass concentration of ambient $PM_{10}$ were in the
order of a few percent estimated by taking into account the relative emission factors of waste burning tracers
determined in controlled laboratory experiments. Considering the extremely high emission factors of PAHs and
their toxicity equivalent from waste burning (Hoffer et al., 2020), this large mass contribution poses
disproportionally higher health risks to the urban population. Using firewood consumption statistics and
emission factors of levoglucosan (a tracer for wood burning,) and the mass concentrations of waste burning
tracers found in atmospheric $PM_{10}$ samples implied that some 10% of all household waste produced end up being
burned in household stoves during the heating season. Albeit these estimates are loaded with very high
uncertainties and can be regarded as back-of-the-envelope calculations only, they prove beyond any doubt that
the illegal burning of solid wastes in household stoves is a common practice in both countries. Our findings call
for the need of immediate and effective legislative measures against these activities posing extreme health risks
to the population in both countries and all over Europe.



**Author contributions.** EAL, LM, AMa collected aerosol samples and performed gravimetric measurements. AH, AT, BJT, AMe GK performed and/or coordinated the analytical measurements and data evaluation. All

authors were involved in the scientific interpretation and discussion of the results as well as in manuscript preparation. All co-authors commented on the paper.

**Competing interests.** The authors declare that they have no conflict of interest.

**Acknowledgement**

This work was supported by the "Analysing the effect of residential solid waste burning on ambient air quality in

central and eastern Europe and potential mitigation measures" (no. 07.027737/2018/788206/SER/ENV.C.3) and by the project RRF-2.3.1-21-2022-00014 of the National Multidisciplinary Laboratory for Climate Change. This work was also supported by the János Bolyai Research Scholarship of the Hungarian Academy of Sciences. The authors would like to thank to Institute for Environment and Energy, Technology and Analytics e.V. (IUTA) for providing the high volume samplers used for PM10 measurements in Romania.

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
