# Peer review of "Assessment of the contribution of residential waste burning to ambient PM10 concentrations in Hungary and Romania"

_EGUsphere, 2023_

## Author Comment (AC1)

**Response to Interactive comment of Anonymous Referee #1**

*Comments and questions of the reviewer are in italics*
Authors' answers are in regular typeface
Parts of the answers highlighted in yellow are inserted into the revised manuscript.

The authors thank the referee for the detailed review and comments. The responses are given below.

*The authors have written a good scientific manuscript with a detailed discussion and analysis. Several minor adjustment should be done in order to increase its quality:*

*1. This manuscript is a continuance study of Hoffer et al. (2021) where used the tracers found in the study for assessing PM10 concentrations by waste burning in several cities of two countries. Therefore, the authors were stated that there has been no study on the estimated mas contribution. However, I found a study which might be similar: https://doi.org/10.1021/acsearthspacechem.2c00243; https://doi.org/10.1016/j.jenvman.2021.112793. Please be sure to highlight the gaps of the written topic*

Certainly, the two papers do share some similarities, particularly the first paper (Islam et al., 2022) regarding the estimation of waste burning's contribution to ambient PM10 levels. Nevertheless, our approach differs from that in the cited papers. Besides the well-known 1,3,5,-triphenylbenzene tracer, we employed multiple and in part specific tracer compounds for various individual waste types in order to assess the waste burning practices in households, as well as the impact of waste burning emission sources on the ambient PM10 concentrations in different settlements. As a result, we have included the following sentence in the introduction to clarify this distinction:

Although there are a few studies on the estimated mass contribution of solid waste burning (Islam et al., 2022) to ambient PM10 concentrations, to the best of our knowledge, our work is the very first study that attempts to quantitatively assess the magnitude of residential waste burning particulate emissions and their contribution to atmospheric PM10 concentrations in different settlements based on the highly specific tracers of waste burning.

*2. Since the focus of this study are the plastic burning, it is also suggested to justify the title directly to the plastic burning in the residential areas.*

Although the majority of findings are indeed related to the combustion of plastics, we have also evaluated the sources related to the burning of fiberboard, scrap furniture, as well as paper, which together make up a sizable mass of all household wastes burned. Therefore we prefer to keep the title of the manuscript as it is.

*3. The habits to burn the waste is also induced by the lack of commitment from government and community as reported here https://doi.org/10.1007/s10163-022-01430-9. Therefore, this*

*information or suggestion can also be added in the introduction (2nd paragraph) or revising the conclusion (for last sentences in the current version)*

The reference by Ramadan et al. (2022) and a sentence have been added to the introduction part.

==Furthermore, the lack of discipline and commitment within the local government and the community might also promote the habit to burn waste outdoors (Ramadan et al., 2022).==

*4. It is suggested to revise the sub section and make a reorganization of the manuscript. One of the problems is on the sub section 2.1. which should be divided into 2 sub section which include the sampling site and PM10 filter sampling.*

The sub section has been divided as suggested into two sub sections:

==2.1 Sampling==

==2.1.1 Sampling sites==

==2.1.2 $PM_{10}$ filter samples==

*5. The sampling location presented in Table 1 should be defined. Even though the acronyms of, for instance, BUD, has been informed in the body text, the authors are suggested to add 1 column to put the meaning of each location acronyms in the table. It is also recommended to add the meaning of each representative location (e.g. rural, residential, background location, road, and so on) in the table.*

Table 1 has been modified: the name and type of the sampling locations have been added to the table. Instead of the RSD, the standard deviations are used to infer the variability of the measured PM10 concentrations.

| Location | Type of the sampling location | Sampling period | Number of samples analyzed | PM$_{10}$ (µg m$^{-3}$) average (SD) |
|---|---|---|---|---|
| BUD (Budapest, Hungary) | urban | 21.01–10.02.2019 | 10 | 58.6 (10.8) |
| | | 01.07–07.07.2019 | 7 | 22.9 (3.8) |
| | | 07.01–29.01.2020 | 21 | 47.4 (16.7) |
| KPS (K-Puszta, Hungary) | rural backround | 21.01–10.02.2019 | 10 | 41.1 (4.7) |
| | | 08.07–14.07.2019 | 7 | 10.0 (2.3) |
| | | 07.01–27.01.2020 | 21 | 39.0 (14.1) |
| MSK (Miskolc, Hungary) | urban | 14.01–03.02.2019 | 10 | 64.6 (12.6) |
| | | 24.06–30.06.2019 | 7 | 23.5 (5.5) |
| | | 07.01–27.01.2020 | 21 | 53.6 (16.3) |
| PUT (Putnok, Hungary) | suburban | 14.01–03.02.2019 | 10 | 81.0 (14.8) |
| | | 24.06–30.06.2019 | 7 | 17.9 (4.7) |
| | | 07.01–27.01.2020 | 21 | 55.8 (24.0) |
| VES (Veszprém, Hungary) | urban | 28.01–17.02.2019 | 10 | 34.8 (8.1) |
| | | 08.07–14.07.2019 | 7 | 10.5 (1.8) |
| | | 07.01–27.01.2020 | 21 | 32.8 (10.6) |
| BUC-R (Bucharest, Romania) | suburban | 22.01-11.02.2019 | 10 | 52.1 (8.7) |
| | | 19.06-25.06.2019 | 7 | 32.4 (3.6) |
| | | 06.02-26.02.2020 | 19 | 36.5 (16.9) |
| BUC-M (Bucharest, Romania) | suburban | - | - | - |
| | | - | - | - |
| | | 06.02-26.02.2020 | 21 | 34.4 (18.1) |
| CLJ (Cluj-Napoca, Romania) | suburban | 26.01-15.02.2019 | 10 | 52.9 (11.9) |
| | | 19.06-25.06.2019 | 7 | 17.2 (4.5) |
| | | 10.01-03.02.2020 | 21 | 39.9 (24.8) |
| DEV (Deva, Romania) | urban | 30.01-19.02.2019 | 10 | 67.8 (11.0) |
| | | 02.07-08.07.2019 | 6 | 28.1 (16.0) |
| | | 08.01-28.01.2020 | 21 | 71.3 (24.3) |
| FOC (Focsani, Romania) | suburban | 19.02-11.03.2019 | 10 | 61.2 (15.3) |
| | | 19.06-25.06.2019 | 7 | 28.0 (3.2) |
| | | 09.01-29.01.2020 | 20 | 49.5 (22.6) |
| | | | Total: 359 | |

**6. The calculation of EF and how the EF can be generated are not presented in the methods section.**

To obtain the average emission factor which was used in the estimation of the contribution of waste burning, both the PM10 emission factor (on a relative scale) reported by Hoffer et al., 2020, and the emission factors of the tracer compounds (both the absolute EFs (mg kg$^{-1}$) and the relative EFs (µg g$^{-1}$ PM10$^{-1}$) reported by Hoffer et al., 2021 were used. Since the emission factors of the tracer compounds depend on the type of the burned material, and a given marker compound might also be

emitted from the burning of different waste types, we performed a sensitivity analysis by calculating the average emission factors for each tracer compounds for different waste mixtures. (For the relative EFs we have calculated the amount of the emitted PM10 and the amount of the given tracer compound in the emitted PM10 for the different waste mixtures, whereas for the absolute EFs given in Table 4 the amount of the emitted tracer compound was related to the weight of the different waste mixtures). As a starting point, we assumed that on a mass basis, the burned household waste consist of 52.6% furniture panels, 15.8% paper, 15.8% rag and 15.8% plastics (the mass ratios of these waste types in this case is 10:3:3:3, respectively), and calculated the resulting average emission factor for each tracer components. We also assumed that the composition of the plastic waste is 42% PE, 28% PET, 14% PP and PS, 0.7% PVC and ABS according to Bodzay and Bánhegyi, (2016). In the calculations we increased the relative amount of the different waste types and/or groups of the waste mixture by a factor of 10. The weighted averages of the emission factors of the marker compounds were calculated for all possible combinations (altogether 15 combinations). Table S1 and table S2 show the obtained EF values for the different waste mixtures. The results of these calculations are summarized in Table 2 which shows the obtained minimum and maximum relative EF values, as well as the average relative EFs of the 15 possible mixture combinations. Table 4 contains the average absolute EF values. Here, we note that the obtained EF values of the different tracers are characteristic only for those waste types the combustion of which produces the given tracer compound (it is not an average EF for the whole waste mixture), as the effects of the different waste burning emission sources are treated separately. (That is why the estimated EF values of melamine which is emitted solely from the burning of furniture panels does not change as the composition of the burned waste varies).

The above description and the obtained EF data for the different waste compositions, which served as the basis for the relative EF data in Table 2, and also for the absolute EF data in Table 4, are presented in the supplementary material.

**7. As it is informed in many part of the manuscript, the uncertainties are really high. However, it is suggested to add the methods for uncertainties assessment in the methods section.**

The manuscript indeed highlights the different sources of uncertainties of the estimations, such as uncertainty of the emission measurements, those resulting from the highly variable burning conditions, degree of co-firing, atmospheric stability of tracers. For the emission factors of the different tracer compounds a simple sensitivity study was performed to address the uncertainty derived from the variability of the composition of the burned household waste (which of course also depends on the initial assumptions), but other parameters are very hard to estimate (e.g. the variability of the result due to the fact that the combustion parameters may also vary both spatially and temporally). On the other hand, the atmospheric mixing processes may average out the differences between individual sources.  To include all these uncertainties whose sources are impossible to assess objectively in the real world was clearly  beyond the scope of the manuscript. That is why the results are presented as order-of-magnitude estimates only regarding the contribution of waste burning to ambient PM10 concentrations.

*8. In the analysis, (L410) the authors mention that the calculation reveals that the household waste are burned 8-15%. How this information can be counted? The average waste generation or information related to waste management in both country should also be stated in the manuscript.*

Yes, indeed to estimate the amount of waste burned in household stoves we used the amount of waste generated as well as the amount of firewood consumed in both countries. The average absolute emission factors and the concentration of tracers in environmental samples can be used to calculate the ratio of waste to biomass burned. Table 3 summarizes these results. Since authorities report the total amount of firewood used in each country, we can calculate the mass of waste burned, which can be compared to the total mass of waste generated in the given country. The data needed for the estimations are given in the manuscript.

*9. How this result can be generated or interpolated in the city context? please briefly explained in the methods*

Of course the applied method might be used to estimate the mass of waste burned in a given settlement, provided that for these calculations the amounts of the waste generated and of the firewood used in the different settlements during the measurement period must be assumed. Since the estimates are already subject to large uncertainties, and these assumptions would certainly add to the uncertainties, the authors did not estimate the amount of waste burned at each locality level, but only provided an estimate at the national level (for which statistical data are only available

**Response to Interactive comment of Anonymous Referee #2**

*Comments and questions of the reviewer are in italics*

Authors' answers are in regular typeface

Parts of the answers highlighted in yellow are inserted into the revised manuscript.

The authors thank the referee for the review and suggestions.

*Paper is well and I do not see very significant issues.*

1. *My biggest concern is related to the table 1. The authors tries to hide enormous standard deviation expressing it as RSD. It is well known that in some cases RSD can be over 100, but, to be honest in your work, you should provide different measure to describe shape of PM concentrations distributions. It expect additional measures as Inter Quartile Range (IQR) or median, quartile 1st and 3rd.*

Table 1 contains the average and the RSD of the ambient PM10 data obtained for the different sampling sites during the various sampling periods. These data give some information on the ambient conditions during the sampling, the average PM10 values indicate the average pollution level of the sampling location, whereas the RSD values indicate the variability of the meteorological conditions and/or that of the source strengths. The standard deviation can be easily calculated from the given RSD and average PM10 concentration values (standard deviation=RSD×averagePM$_{10}$/100).

Since the median and IQR (quartile $1^{st}$ – quartile $3^{rd}$) would give information on half of the obtained values, the authors keep the reported average PM10 data in the table but instead of the RSD the standard deviations are shown.

2. *Figures 2-4 have missing bar - exclude tick on axis or add missing bar. If it was intended to show different countries, it failed.*

The misleading tick has been removed from all the figures as recommended.

3. *During the reading of Tables 2 and 3 I found very significant concern - what are uncertainties of these estimates? Did you even included any uncertainty assessment of estimates?*

The manuscript indeed highlights the different sources of uncertainties of the estimations, such as uncertainty of the emission measurements, those resulting from the highly variable burning conditions, degree of co-firing, atmospheric stability of tracers. For the emission factors of the different tracer compounds a simple sensitivity study was performed to address the uncertainty derived from the variability of the composition of the burned household waste (which of course also depends on the initial assumptions), but other parameters are very hard to estimate (e.g. the variability of the result due to the fact that the combustion parameters may also vary both spatially and temporally). On the other hand, the atmospheric mixing processes may average out the

differences between individual sources. To include all these uncertainties whose sources are impossible to assess objectively in the real world was clearly beyond the scope of the manuscript. That is why the results are presented as order-of-magnitude estimates only regarding the contribution of waste burning to ambient PM10 concentrations.

.

4. ***In conclusion, although I see that your study is important and results are good and interesting, you should slightly more underline what you achieved.***

To underline the achieved results, the following sentences have been modified:

It is very important to emphasize that considering the extremely high emission factors of PAHs and their toxicity equivalent from waste burning (Hoffer et al., 2020), this large mass contribution poses disproportionally higher health risks to the urban population.

Albeit these estimates are loaded with very high uncertainties and can be regarded as back-of-the-envelope calculations only, it is crucial to underline that they prove beyond any doubt that the illegal burning of solid wastes in household stoves is a common practice in both countries.